# [^99^mTc]Sestamibi SPECT Can Predict Proliferation Index, Angiogenesis, and Vascular Invasion in Parathyroid Patients: A Retrospective Study

**DOI:** 10.3390/jcm9072213

**Published:** 2020-07-13

**Authors:** Nicoletta Urbano, Manuel Scimeca, Carmela Di Russo, Alessandro Mauriello, Elena Bonanno, Orazio Schillaci

**Affiliations:** 1Nuclear Medicine Unit, Department of Oncohaematology, Policlinico “Tor Vergata”, 00133 Rome, Italy; n.urbano@virgilio.it (N.U.); carmela.dirusso@ptvonline.it (C.D.R.); 2Department of Biomedicine and Prevention, University of Rome “Tor Vergata”, Via Montpellier 1, 00133 Rome, Italy; manuel.scimeca@uniroma2.it; 3San Raffaele University, Via di Val Cannuta 247, 00166 Rome, Italy; 4Saint Camillus International University of Health Sciences, Via di Sant’Alessandro, 8, 00131 Rome, Italy; 5Department of Experimental Medicine, University of Rome “Tor Vergata”, Via Montpellier 1, 00133 Rome, Italy; alessandro.mauriello@uniroma2.it (A.M.); elena.bonanno@uniroma2.it (E.B.); 6Diagnostica Medica’ & ‘Villa dei Platani’, Neuromed Group, 83100 Avellino, Italy; 7IRCCS Neuromed, Via Atinense, 18, 8607 Pozzilli, Italy

**Keywords:** hyperparathyroidism, parathyroid cancer, [^99^mTc]Sestamibi SPECT, proliferation index, angiogenesis

## Abstract

The aim of this study was to evaluate the possible association among sestamibi uptake and the main histopathological characteristics of parathyroid lesions related to aggressiveness such as the proliferation index (Ki67 expression and mitosis), angiogenesis (number of vessels), and vascular invasion in hyperparathyroidism patients. To this end, 26 patients affected by primary hyperparathyroidism subjected to both scintigraphy with [^99^mTc]Sestamibi and surgery/bioptic procedure were retrospectively enrolled. Hyperfunctioning of the parathyroid was detected in 19 patients. Our data showed a significant positive association among the sestamibi uptake and the proliferation index histologically evaluated both in terms of the number of Ki67 positive cells and mitosis. According to these data, lesions with a higher valuer of L/N (lesion to nonlesion ratio) frequently showed several vessels in tumor areas and histological evidence of vascular invasion. It is noteworthy that among patients with negative scintigraphy, 2 patients showed a neoplastic lesion after surgery (histological analysis). However, it is important to highlight that these lesions displayed very low proliferation indexes, which was evaluated in terms of number of both mitosis and Ki67-positive cells, some/rare vessels in the main lesion, and no evidence of vascular invasion. In conclusion, data obtained on patients with positive or negative scintigraphy support the hypothesis that sestamibi can be a tracer that is capable of predicting some biological characteristics of parathyroid tumors such as angiogenesis, proliferation indexes, and the invasion of surrounding tissues or vessels.

## 1. Introduction

Recent data indicate a continuous increase in hyperparathyroidism (PHP) incidence [1]. It affects the patient’s quality of life by dysregulating calcium homeostasis and thereby inducing various multiple organ complications. Currently, surgery represents the unique definite treatment for PHP patients [1].

Most cases of primary PHP are related to the presence of (1) solitary parathyroid adenoma (85% of cases), (2) glandular hyperplasia (10%), (3) multiple adenomas (4%), and (4) parathyroid carcinoma (<1%) [1]. Moreover, epidemiological data indicate that the sporadic incidence of ectopic parathyroid adenomas is 5–16% [1]. Histologically, these adenomas arise mainly from chief cells, although some adenomas can show a mixture of oxyphil cells and transitional oxyphil cells [2]. Numerous histological variants of parathyroid adenomas such as lipoadenoma, cystic adenoma, and oncocytic adenoma have been described [3]. Despite parathyroid carcinoma being described as an uncommon malignancy, its incidence has significantly increased in PHP patients, especially in the last few years [4]. Generally, parathyroid carcinoma is larger than adenoma. Microscopically, it appears lobulated, firm, and attached to surrounding soft tissue structures [5]. Patients affected by parathyroid carcinoma typically show high PTH and calcium serum levels also presenting severe and atypical clinical picture [2]. The dysregulation of both PTH and calcium homeostasis frequently affect the norm physiology of bone and kidney [2].

Several studies investigated the carcinogenesis of parathyroid carcinoma, as well as its relationship with PHP [6,7,8]. However, both molecular mechanisms and clinical features of parathyroid carcinomas are not yet fully understood.

The parathyroid glands are able to concentrate a variety of chemical substances, including vital dyes and radiopharmaceuticals, and this ability has been exploited for localization purposes. In this scenario, over the years, numerous methods of performing parathyroid imaging with sestamibi have been suggested [9,10,11,12,13].

Specifically, parathyroid scintigraphy by [^99^mTc]Sestamibi Single Photon Emission Computed Tomography (SPECT) is considered a conventional analysis to detect abnormal parathyroid glands [9]. However, it is important to note that parathyroid imaging rarely has role in the diagnosis of PHP, which is based on biochemical and clinical data but plays an essential role in the preoperative identification of hyperfunctioning parathyroid gland(s). Indeed, parathyroid scintigraphy is generally used in the management of PHP patients as an imaging procedure that is capable of providing important information to schedule a minimally invasive parathyroidectomy [10,11]. However, recent discoveries about the ability of [^99^mTc]Sestamibi SPECT to identify some biological characteristics of human carcinomas, i.e., breast cancers [12,13], can provide a scientific rationale to study the possible association between sestamibi uptake and histological characteristics of parathyroid lesions. In particular, the uptake of sestamibi in the mitochondria [14] could be related to the aggressiveness of parathyroid cancers, which histologically can appear as a high proliferation index, the presence of numerous intratumoral vessels (angiogenesis), and vascular invasion. Nevertheless, to the best of our knowledge, no study tested this hypothesis.

Starting from these considerations, the aim of this study was to evaluate the possible association among sestamibi uptake and the main histopathological characteristics of parathyroid lesions related to aggressiveness such as proliferation index (Ki67 expression and mitosis), angiogenesis (number of vessels), and vascular invasion in PHP patients. To this end, [^99^mTc]Sestamibi SPECT and histological data (both morphology and immunohistochemistry) from 26 PHP patients have been collected.

## 2. Patients and Methods

The “Policlinico Tor Vergata” Ethical Committee approved this protocol with the reference number #129.18. In addition, all methodologies and experimental procedures here described were achieved in agreement with the last Helsinki Declaration.

Exclusion criteria: a second cancer and neoadjuvant hormonal or radiation therapy prior to surgery. According to these criteria, we retrospectively enrolled 26 consecutive patients with parathyroid dysfunction (64.81 ± 2.56 years; range 33–82 years; 20 women and 6 men), who underwent both [^99^mTc]Tc-SPECT with Sestamibi and a parathyroid bioptic procedure from January 2018 to December 2019. For each of them, histological diagnosis and immunohistochemical investigations were performed.

### 2.1. [^99^mTc]Sestamibi SPECT

Early acquisition was performed at 15 min after the intravenous injection of 740 MBq Tc-sestamibi (Bristol-Myers Squibb Pharma, Bruxelles, Belgium) according to the recommendations of the European Association of Nuclear Medicine [15]. Planar images of the neck and chestwere were obtained in a 256 × 256 matrix, with a 20% energy window centered at a 140 keV photopeak using a high-resolution SPECT system (Millenium VG & Hawkeye; General Electric Medical Systems, Milwaukee, WI, USA) equipped with low-energy high-resolution parallel-hole collimators. Patients were positioned supine, with the neck supported in an extended position and arms lowered alongside the body. A step-and-shoot protocol was used that consisted of 20 s per frame with a total of 64 frames. Transverse, coronal, and sagittal SPECT images were generated by using a Gaussian 2.0 prefilter, and they were post-processed by using fast low-angle shot three-dimensional iterative reconstruction (four iterations, eight subsets). An attenuation correction factor of 0.15/cm was applied with the Chang method.

All 26 patients had biopsy. [^99^mTc]Sestamibi SPECT was performed before biopsy in 15 patients and after biopsy in 11 patients. When [^99^mTc]Sestamibi SPECT was performed after biopsy, the minimum interval between biopsy and imaging was 7 days in an effort to avoid the effects of post-biopsy inflammation as much as possible.

For qualitative analysis of [^99^mTc]Sestamibi SPECT, two investigators classified positive and negative findings. Lesions with no demonstrable uptake and those with diffuse heterogeneous or minimal patchy uptake were considered negative, whereas lesions with scattered patchy uptake, partially focal uptake, or any other focal uptake were regarded as positive. Irregular-shaped regions of interest (ROIs) were used to encase the lesions. The evaluation of the lesion to nonlesion ratio (L/N) was estimated according to our previous study [12]. For the patients who underwent [^99^mTc]Sestamibi SPECT before biopsy (n = 15), the [^99^mTc]Sestamibi SPECT-guided biopsy procedure was performed. Semiquantitative analysis of the [^99^mTc]Sestamibi was performed.

### 2.2. Histology

Parathyroid bioptic samples were formalin fixed and embedded in paraffin [16]. Serial sections were used for both hematoxylin–eosin (H&E) and immunohistochemicalstaining for Ki67. For each sample, three H&E serial sections were used to evaluate the number of mitosis and the number of vessels on 10 High Power Field (HPF; 40 × magnification) randomly selected cancer areas. In all sections, the vascular and cap invasion was also assessed. All morphological evaluations were performed by using digital slides (Iscan Coreo, Ventana, Tucson, AZ, USA).

### 2.3. Immunohistochemistry

Immunohistochemistry was used to study the proliferation index by Ki67 expression. Three-μm-thick paraffin sections were treated with Citrate buffers pH 6.0 for 30 min at 95 °C to antigen retrieval reaction. Afterwards, sections were incubated with pre-diluted anti-Ki67 rabbit monoclonal antibody (clone 30-9, Ventana, Tucson, AZ, USA). Washings were performed with PBS/Tween20 pH 7.6. Reactions were detected by using an HRP-DAB Detection Kit (UCS Diagnostic, Rome, Italy).

A digital scan was used to evaluate the immunohistochemical reactions (Iscan Coreo, Ventana, Tucson, AZ, USA). Specifically, Ki67 was calculated in terms of percentage of positive parathyroid cells. Reactions have been set up by using specific positive and negative control tissues. Specifically, negative controls were perfomed on serial paraffin section without using primary antibody, wherear positive controls were performed by investigated the Ki67 expression on thymus paraffin sections.

### 2.4. Statistical Analysis

In order to evaluate the possible association among sestamibi uptake, the age, percentage of Ki67 positive cancer cells, number of mitosis, number of vessels, and vascular invasion linear regression analyses were performed. One-way ANOVA was used to evaluate the L/N ratio in parathyroid histotypes (3 groups). The difference between groups was considered statistically significant at *p* < 0.05.

## 3. Results

### 3.1. [^99^mTc]Sestamibi SPECT Analysis

[^99^mTc]Sestamibi SPECT analyses showed sestamibi uptake in 19 patients (L/N max 2.78; min 0.85) (Figure 1A). Conversely, no sestamibi uptake was observed in 7 patients (Figure 1B). No significant differences were observed by comparing L/N ratio and parathyroid histotypes.

### 3.2. Histology

Parathyroid biopsies were classified according to the World Health Organization [17]. In particular, we found 8/26 hyperplasia, 8/26 parathyroid adenoma, and 10/26 parathyroid carcinoma. No secondary, mesenchymal, and other tumors were observed. Interestingly, parathyroid tumors (1 parathyroid adenoma and 1 parathyroid carcinoma) were detected in 2 patients with no sestamibi uptake. No association was found between sestamibi uptake and parathyroid histotypes (hyperplasia L/N 1.62 ± 0.36; parathyroid adenoma L/N 1.85 ± 0.84; parathyroid carcinoma L/N 2.02 ± 1.29; *p* = 0.678).

### 3.3. Sestamibi Uptake vs. Cancer Cells Proliferation

To investigate the possible association between sestamibi uptake and cells proliferation in parathyroid lesions, linear regression analyses were performed (Figure 2). Interestingly, positive significant associations were found by comparing the L/N ratio with both Ki67 index (*p* = 0.0003; r^2^ 0.4657) and the number of mitosis (*p* = 0.0002; r^2^ 0.4720) (Figure 2A,B,E–J).

It is important to note the high concordance between the value of mitosis and the percentage of Ki67 positive cells.

To exclude the influence of age on both sestamibi uptake and proliferation index, linear regression analyses were performed between age and both L/N ratio and Ki67 value (Figure 2C,D). Of note, no positive associations were found (year versus L/N *p* = 0.4465 r^2^ 0.0278; year versus Ki67 *p* = 0.5513; r^2^ 0.0149) (Figure 2C,D). It is noteworthy that both parathyroid adenoma and parathyroid carcinoma of patients with negative [^99^mTc]Sestamibi SPECT showed very low values of Ki67 and number of mitosis.

### 3.4. Sestamibi Uptake, Angiogenesis, and Vascular Invasion

Linear regression analysis was performed to study the possible association between sestamibi uptake and the number of vessels in the tumor area. It is noteworthy that a positive significant association was observed (*p* = 0.0148 r^2^ 0.2513) (Figure 3). In order to establish the capability of [^99^mTc]Sestamibi SPECT analyses to predict the aggressiveness of parathyroid carcinomas, we subdivided selected lesions according to the presence of vascular invasion, which was evaluated in terms of the presence of cancer cells in at least 2 vessels. Our data showed a significant increase in sestamibi uptake in lesions characterized by vascular invasion as compared to lesions without any histological evidence of vascular invasion (*p* = 0.0377) (Figure 3). It is important to note that biopsies of patients affected by hyperplasia have been excluded from the analyses of vessels and vascular invasion. Of interest, no vascular invasion was observed in both parathyroid adenoma and parathyroid carcinoma of patients with negative [^99^mTc]Sestamibi SPECT.

## 4. Discussion

PHP represents the most common disorder of the endocrine system, with a prevalence of up to 1% and increased incidence in women and with advanced age [1] Clinical studies reported several co-morbidities related to PHP such as musculoskeletal, neuropsychiatric, gastrointestinal, renal, and cardiovascular disorders. Thus, the occurrence of PHP is associated to both a significant reduction of the patient’s quality of life and an increase of risk for morbidity [18]. In addition, several studies reported an increase of the risk of parathyroid carcinoma occurrence in patients affected by PHP [19,20]. Indeed, despite parathyroid carcinoma being described as an uncommon malignancy, its incidence significantly increases in patients affected by PHP [20]. Nevertheless, the pathogenesis of parathyroid carcinoma is not fully understood yet. Therefore, the diagnosis of these tumors is considered a diagnostic challenge due to the absence of peculiar characteristics that allow a definite distinction of malignant from benign disease. Concerning the therapy, currently, surgery remains the only curative approach for both PHP and parathyroid carcinoma, also allowing the identification of the histological and molecular characteristics of these lesions.

From a diagnostic point of view, parathyroid scintigraphy is often used to detect a hyperfunctioning parathyroid tissue in patients with PHP prior to surgery [10,21,22]. In this context, [^99^mTc]Sestamibi is the main radiotracer employed in parathyroid scintigraphy, since this molecule remains longer in the mitochondria of the parathyroid rather than thyroid, where it is washed out quickly [23,24].

Despite this, the predictive role of sestamibi uptake in the occurrence and progression of parathyroid lesions, as well as the association with histopathological characteristics, represents an open question in the management of patients affected by PHP and/or parathyroid carcinoma.

Starting from these considerations, the aim of this study was to evaluate the possible association among sestamibi uptake and the main histopathological characteristics of parathyroid lesions related to aggressiveness such as proliferation index (Ki67 expression and mitosis), angiogenesis (number of vessels), and vascular invasion in PHP patients.

To this end, 26 PHP patients subjected to both scintigraphy with [^99^mTc]Sestamibi and surgery/bioptic procedure were retrospectively enrolled. Hyperfunctioning of the parathyroid was detected in 19 patients.

Our data showed a significant positive association among the sestamibi uptake and the proliferation index evaluated both in terms of the number of Ki67 positive cells and mitosis. This can be explained by (1) the capability of sestamibi to remain in the mitochondria after passive diffusion [14], (2) the increased uptake of sestamibi in mitochondria with high membrane potential [12], and (3) the role of mitochondria and their membrane potential in the cell proliferation process [25]. Of note, we also found that age affects neither the sestamibi uptake nor proliferation index in our case selection.

According to data of the association between proliferation index and sestamibi uptake, we observed that lesions with a higher L/N value (lesion to nonlesion ratio) frequently showed several vessels in tumor areas and histological evidence of vascular invasion. Therefore, the uptake of sestamibi increased in metabolically active lesions characterized by tumors cells proliferation, angiogenesis, and invasion. These characteristics are strongly related to the capability of tumors to grow and invade surrounding tissues [26,27]. In particular, the angiogenesis phenomenon has been associated to metastatic spread in parathyroid cancers by Garcia de la Torre and colleagues [26].

It is noteworthy that among patients with negative scintigraphy, 2 patients showed a neoplastic lesion after surgery (histological analysis). However, it is important to highlight that these lesions displayed very low proliferation indexes, which were evaluated in terms of the number of both mitosis and Ki67 positive cells, some/rare vessels in the main lesion, and no evidence of vascular invasion. Thus, our data seem to indicate that scintigraphy with [^99^mTc]Sestamibi could underestimate the presence of parathyroid lesions with a greater morphological aspect of neoplasia but low aggressiveness. Nichols et al. demonstrated that most frequently, false negative scintigraphy with [^99^mTc]Sestamibi occurs in the presence of parathyroid lesions contiguous with the upper or lower poles of the thyroid gland [28]. However, to the best of our knowledge, no study investigated the possible association between negative scintigraphy with [^99^mTc]Sestamibi and histopathological characteristics of parathyroid lesions. In this study, for the first time, a relationship between sestamibi uptake and the histopathological characteristics of parathyroid tumors was shown. Both data obtained on patients with positive and negative scintigraphy support the hypothesis that sestamibi can be a tracer that is capable of predicting some biological characteristics of parathyroid tumors such as angiogenesis, proliferation indexes, and the invasion of surrounding tissues or vessels. The possibility of detecting these characteristics by in vivo analysis opens new perspectives in the management of PHP patients. Indeed, our data, if confirmed on a large cohort of patients, could be used to develop diagnostic protocols that are capable of stratifying PHP patients according to prognostic and predictive information generally provided by histological and immunohistochemical analysis. In addition, our approach can be used for other diseases, thus expanding the diagnostic “equipment” available to nuclear physicians.

## 5. Conclusions

The capability of sestamibi to identify malignant lesions by SPECT analysis has been shown for several human diseases such as hearth injury, breast cancer renal carcinomas, and PHP [29,30,31,32,33]. Indeed, numerous investigations demonstrated a close association between positive [^99^mTc]Sestamibi analysis and the severity of disease [29,30,31,32,33]. However, few studies correlated the sestamibi uptake with the histopathological characteristics of human lesions [34,35]. Therefore, despite preliminary findings, the results of this study can support the physicians in the evaluation of [^99^mTc]Sestamibi SPECT in PHP patients. In general, the association between nuclear medicine and anatomic pathology data could provide the scientific rationale for developing new in vivo diagnostic methods that are capable of predicting prognosis or response to therapy for human cancers.

## Figures and Tables

**Figure 1 jcm-09-02213-f001:**
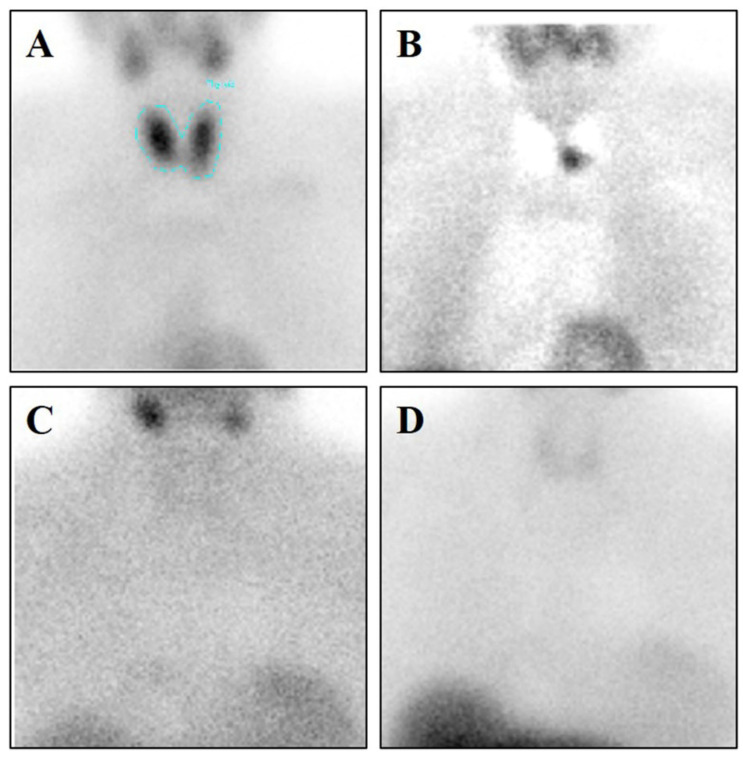
[^99^mTc]Sestamibi Single Photon Emission Computed Tomography (SPECT) Analysis. (**A**) Image shows [^99^mTc]Sestamibi uptake in a 54-year-old woman with primary hyperparathyroidism. A parathyroid carcinoma (0.6 cm) was identified after the surgery by histological analysis. (**B**) To evaluate the parathyroid sestamibi uptake, that of the thyroid has been subtracted (**C**) Image displays no [^99^mTc]Sestamibi uptake in a 68-year-old woman with primary hyperparathyroidism. A parathyroid hyperplasia (0.2 cm) was identified after the surgery by histological analysis. (**D**) To evaluate the parathyroid sestamibi uptake, that of the thyroid has been subtracted.

**Figure 2 jcm-09-02213-f002:**
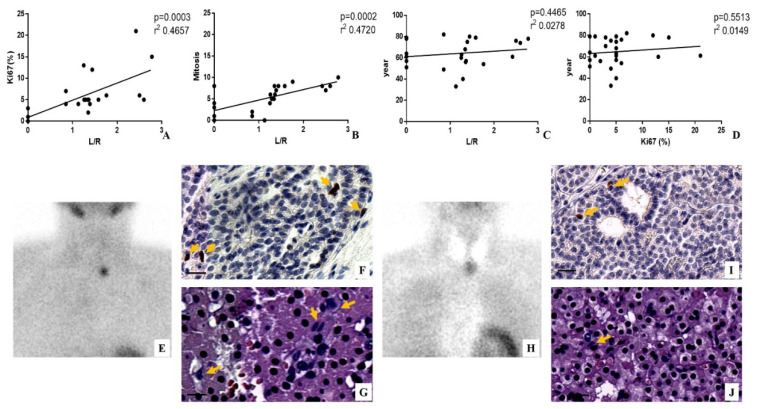
Evaluation of sestamibi uptake and proliferation index in patients affected by hyperparathyroidism. (**A**) Graph shows linear regression analysis between the percentage of Ki67 positive cells and lesion to nonlesion (L/N) ratio. (**B**) Graph displays linear regression analysis between the number of mitosis and L/N ratio. (**C**) Graph shows linear regression analysis between patients’ year and L/N ratio. (**D**) Graph displays linear regression analysis between patients’ year and the percentage of Ki67-positive cells. (**E**) Image shows [^99^mTc]Sestamibi uptake in a 74-year-old woman with primary hyperparathyroidism. A parathyroid carcinoma (0.8 cm) was identified after the surgery by histological analysis. To evaluate the parathyroid sestamibi uptake, that of the thyroid has been subtracted. (**F**) Representative image of immunohistochemical reaction for ki67. Several positive Ki67 cancer cells are labeled by arrows. (**G**) Hematoxylin–eosin (H&E) staining shows several mitosis (arrows). (**H**) Image displays no [^99^mTc]Sestamibi uptake in a 40-year-old woman with primary hyperparathyroidism. A parathyroid adenoma (0.4 cm) was identified after the surgery by histological analysis. (**I**) Representative image of immunohistochemical reaction for Ki67. Rare positive Ki67 cancer cells are labeled by arrows. (**J**) H&E staining shows rare mitosis (arrows). Scale bar represents 100 µm in all images.

**Figure 3 jcm-09-02213-f003:**
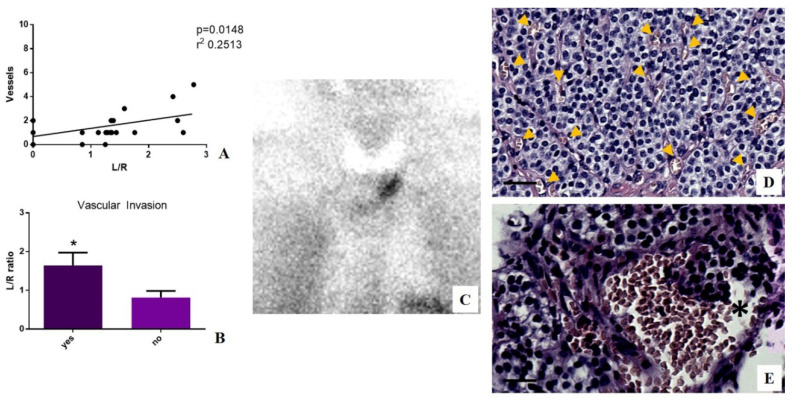
Evaluation of sestamibi uptake, angiogenesis, and vascular invasion in patients affected by hyperparathyroidism. (**A**) Graph shows linear regression analysis between the number of vessels and L/N ratio. (**B**) Graph displays a significant increase in sestamibi uptake in parathyroid lesions showing vascular invasion (yes) as compared to lesions without vascular invasion (no) (* *p* = 0.0377). (**C**) Image shows [^99^mTc]Sestamibi uptake in a 61-year-old man with primary hyperparathyroidism. A parathyroid carcinoma (0.7 cm) was identified after the surgery by histological analysis. (**D**) Image displays numerous vessels (yellow arrows) in a tumor area. (**E**) Image shows a large vessel with several cancer cells (asterisk). Scale bar represents 100 µm in all images.

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
