# Peer review of "[99mTc]Sestamibi SPECT Can Predict Proliferation Index, Angiogenesis, and Vascular Invasion in Parathyroid Patients: A Retrospective Study"

_jcm, 2020, doi:10.3390/jcm9072213_

Round 1

Reviewer 1 Report

Page 1, line 21 and also elsewhere (p.2, lines: 58,65,87, p.3, line 103, etc. ) in the text sould be corectly stated [99mTc]Sestamibi (aslo with upper index) instead of [99mTc]Tc-Sestamibi, which is not correct according IUPAC nomeclature. 

Page 2, line 92 and 93, also elewhere in the tex is listed [99mTc]Tc-SPECT, listed term is not appropriate. More eligible is Sestamibi SPECT, or simply SPECT or SPECT with /using [99mTc]Sestamibi or alternatively [99mTc]Sestamibi SPECT.

Page 2, line 87: Reconstitution/preparation of [99mTc]Sestamibi is probably not described clearly. Synthesis of labelled [99mTc]Sestamibi was certainly performed on site at the dpt. of nucler medicine with eluate from 99Mo/99mTc generator. Also the type of generator should be listed. Radiochemical/radionuclide purrity of [99mTc]Sestamibi should be mentioned. Authors should also explain why they apply activity 740 MBq. Was the same dose 740 MBq administred to all patients or if the administered activities in patient cohort were corrected on weight of patients? How long was acquisition time?

Page 1, line 25 vs. p. 3, line 101: Please, explaine the difference between L/R ratio and L/N ratio? In the text, both terms (L/R and L/N) are used in the same sense. Could you explain it? Even though authors did not observe any significant differences between L/N ratios and parathyroid histo-types the data might be given for completeness. 

Page 4, Figure 1: The description of pictures is difficult to understand. Could you link each picture with the proper description?

Page 5, Figure 2: Arrows indicate the presence of leasion in the picture are not visible. They merge with pictures. The same situation repeat on Figure 3, page 6 (asterix). In both figures it should be fixed. 

Page 5, line 167 and 171: There shlould be Ki67 instead of ki67.

The conclusion presented are not in the line with the objectives of work. There are no main conlusion about hte role of [99mTc]Sestamibi in scintigraphy of PHP patiens.

Reviewer 2 Report

  1. The authors might elaborate further on the relationship between sestamibi uptake and malignacy. There is much more then their own work.
  2. Since hyperparathyroidisme is a common sign in patients with parathyroid carcinoma it is not uncommon to find a higher incidence in such a group, like the incidence of breast cancer in patients with a palpable mass.
  3. The scintigraphy could be explained in more detail. What approach was used for subtraction? Sodium -[123 I]I? Or wash-out? What were the activities of the tracers used? What was the timing of the image acquisition?
  4. Do the authors see a role for sestamibi scintigraphy in predicting the presence of a carcinoma vs adenoma? Why or why not.
  5. Did the authors observe a relation between sestamibi uptake and plasma concentration of parathyroid hormone?
  6. In line 146 the authors mention an association between sestamibi uptake and breast cancer cell proliferation, maybe parathyroid carcinoma was meant?
  7. L/N and L/R are used for the same target to background measurement? The authors might choose more common TBR.
  8. Use [99mTc]Tc-sestamibi-SPECT (sorry superscript of 99m does not work in this text editor.) Also see https://www.eanm.org/publications/guidelines/nomenclature/
